# Gender Differences in Knowledge and Perception of Cardiovascular Disease among Italian Thalassemia Major Patients

**DOI:** 10.3390/jcm11133736

**Published:** 2022-06-28

**Authors:** Antonella Meloni, Laura Pistoia, Silvia Maffei, Paolo Marcheschi, Tommaso Casini, Anna Spasiano, Pier Paolo Bitti, Liana Cuccia, Elisabetta Corigliano, Paola Maria Grazia Sanna, Francesco Massei, Vincenzo Positano, Filippo Cademartiri

**Affiliations:** 1Department of Radiology, Fondazione G. Monasterio CNR-Regione Toscana, 56124 Pisa, Italy; antonella.meloni@ftgm.it (A.M.); laura.pistoia@ftgm.it (L.P.); positano@ftgm.it (V.P.); 2Unità Operativa Complessa di Bioingegneria, Fondazione G. Monasterio CNR-Regione Toscana, 56124 Pisa, Italy; 3Cardiovascular and Gynaecological Endocrinology Unit, Fondazione G. Monasterio CNR-Regione Toscana, 56124 Pisa, Italy; silvia.maffei@ftgm.it; 4Reparto INFOTEL, Fondazione G Monasterio CNR-Regione Toscana, 56124 Pisa, Italy; marcheschi@ftgm.it; 5Centro Talassemie ed Emoglobinopatie, Ospedale “Meyer”, 50132 Firenze, Italy; tommaso.casini@meyer.it; 6Unità Operativa Semplice Dipartimentale Malattie Rare del Globulo Rosso, Azienda Ospedaliera di Rilievo Nazionale “A. Cardarelli”, 80131 Napoli, Italy; spasiano.anna@tiscali.it; 7Servizio Immunoematologia e Medicina Trasfusionale, Dipartimento dei Servizi, Presidio Ospedaliero “San Francesco” ASL Nuoro, 08100 Nuoro, Italy; centrotrasfusionale.hsfnuoro@atssardegna.it; 8Unità Operativa Complessa Ematologia con Talassemia, Azienda di Rilievo Nazionale ad Alta Specializzazione Civico “Benfratelli-Di Cristina”, 90127 Palermo, Italy; liana.cuccia@arnascivico.it; 9Ematologia Microcitemia, Ospedale San Giovanni di Dio, Azienda Sanitaria Provinciale di Crotone, 88900 Crotone, Italy; krthal@libero.it; 10Servizio Trasfusionale Aziendale, Azienda Ospedaliero Universitaria di Sassari, 07100 Sassari, Italy; paola.sanna@aousassari.it; 11Unità Operativa Oncoematologia Pediatrica, Azienda Ospedaliero Universitaria Pisana, Stabilimento S. Chiara, 56100 Pisa, Italy; f.massei@med.unipi.it

**Keywords:** awareness, knowledge, gender differences, cardiovascular risk, thalassemia major

## Abstract

We evaluated gender differences in knowledge and perception of cardiovascular disease (CVD) among Italian thalassemia major (TM) patients. An anonymous questionnaire was completed by 139 β-TM patients (87 (62.7%) females, 40.90 ± 8.03 years). Compared to females, males showed a significantly higher frequency of CVDs, and they less frequently selected tumors in general as the greatest health problem for people of the same age and gender (48.1% vs. 66.7%; *p* = 0.031) and as the greatest danger to their future health (26.9% vs. 43.7%; *p* = 0.048). CVDs were designated as the greatest danger to their future health by a significantly higher percentage of males than females (53.8% vs. 36.8%; *p* = 0.048). Both males and females showed a good knowledge of cardiovascular risk factors and preventive measures for CVDs. No gender differences were detected in the subjective well-being and the perceived cardiovascular risk. The perceived risk was not influenced by age, presence of cardiovascular risk factors, or disease, but no patient with a low perceived CVD risk had myocardial iron overload. Our findings highlight the need to implement future educational programs aimed at increasing the awareness of CVD as the greatest health issue, especially among the female TM population, and at informing TM patients of the different actors, besides iron, that play a role in the development of cardiovascular complications.

## 1. Introduction

Beta-thalassemia major (β-TM) is an inherited disease characterized by the absence or severe deficiency of β-globin chain synthesis. TM patients require lifelong regular transfusions to correct anemia and suppress the high level of ineffective erythropoiesis [1]. Due to the absence of an active mechanism for iron excretion [2], the leading culprit of this lifesaving procedure is the accumulation of iron in the liver, endocrine glands, and heart, with consequent organ dysfunction and damage [2]. Iron-induced cardiomyopathy remains the main cause of mortality [3,4,5], despite significant improvement in the survival curve, driven by both the introduction of oral iron chelators (Deferiprone and Deferasirox) [6] and the deployment of the T2* magnetic resonance imaging (MRI) for the non-invasive assessment of myocardial iron overload (MIO) [7]. Indeed, the T2* MRI has permitted the design of tailor-made chelation therapies customized for each patient and the evaluation of their efficacy [8,9], with a consequent reduction of the cardiac iron burden.

Besides MIO, in TM, heart disease can also result from other causes such as chronic anemia, endocrine abnormalities [10,11], nutritional deficiencies, and genetic factors [12,13]. It has been shown that in TM diabetes mellitus significantly increases the risk for heart failure (HF), hyperkinetic arrhythmias, and myocardial fibrosis, independently from MIO [11]. Of note, with improved control of MIO, myocardial fibrosis has arisen as the strongest predictor of heart failure and cardiac complications [14]. In adult TM, traditional cardiovascular risk factors (CVRFs) seem to be implicated in the pathogenesis of myocardial scarring [12]. In regards to chronic hemolytic anemia, besides causing the well-known enlargement of cardiac chambers [15,16], it might also be a predisposing factor for atherosclerosis by several mechanisms, including the increased risk of hypertriglyceridemia [17,18]. All these findings seem to suggest that tight control of modifiable CVRFs may help to further reduce the cardiovascular risk in TM, especially in view of the increased time span to play their role offered by the increasing life expectancy.

Most risk factors can be altered or controlled with lifestyle changes but this requires their recognition. Indeed, general knowledge and awareness of cardiovascular diseases (CVDs) and CVRFs are crucial for making lifestyle changes, adopting preventive measures, and complying with treatment recommendations [19]. Moreover, there is a link between awareness of personal risk and adoption of secondary prevention measures [20].

Very few studies have addressed population awareness of CVDs in Europe, and a recent survey conducted on almost 5000 Italian women without hematological diseases showed that about 20% of women recognized the CVD as their greatest health problem, while tumors in general were cited in more than half of the cases [21]. Moreover, Italian women showed an awareness of CVDs as the leading cause of death (69.8%) higher than American (56%) [22] and Australian (32%) [23] women, and a good knowledge of the major CVRFs.

The knowledge, perception, and awareness of CVRFs and CVDs among TM patients have never been explored and, as a result, it is not known whether in this population the gender affects CVD or risk factor awareness and self-assessment of personal risk. Female patients with TM have a better prognosis than males, mainly due to a lower prevalence of cardiac disease in females [3,24,25]. Importantly, it has been shown that the higher risk of ventricular dysfunction, heart failure, arrhythmias, and cardiac complications detected in the male population is not paralleled by a greater iron accumulation in the heart [26,27].

Based on this background, the aim of this study was to evaluate the gender differences in knowledge and perception of cardiovascular disease among Italian thalassemia major patients.

## 2. Materials and Methods

### 2.1. Study Design and Subjects

We considered 139 β-TM patients (87 (62.7%) females, 40.90 ± 8.03 years), consecutively enrolled in the Extension-Myocardial Iron Overload in Thalassemia (E-MIOT) project, an Italian network constituted by 66 thalassemia centers and 11 validated MRI sites [28,29,30], linked by a shared database, collecting all anamnestic, clinical, and instrumental data. Patients attending the reference MRI center of the E-MIOT Network (Pisa) were approached and asked to participate in the cross-sectional survey. All patients agreed to participate in the study (response rate 100%) and they had the opportunity to complete the survey on-site or to take the survey home and return it by email.

The study complied with the Declaration of Helsinki and was approved by the ethical committee. All patients gave written informed consent.

### 2.2. Survey

The questionnaire was previously validated in the general population [21]. It included three main parts.

Part 1, “General Evaluation”, collected general information about the survey participants, as well as about their knowledge of CVD as the biggest health problem or the greatest health risk.

Part 2, “Cardiovascular Disease”, investigated the knowledge of traditional cardiovascular risk factors and preventive measures. Patients were asked if they were aware on the fact that a given condition/behavior represented a CVRF and that a specific behavior or action could lower the risk of heart disease.

Part 3, “Individual Health Status”, collected data about the perception of subjective health status and personal CVD risk profile. Self-Rating of Health (SRH) was measured by a single question “How would you rate your current health status on a scale from 0 to 10?”. Ratings for SRH were poor <4, acceptable 5–7, and good 8–10. To evaluate the perceived risk of heart disease, patients were asked to answer the question “Which is your personal risk of developing a cardiovascular disease in the future on a scale from 0 (really low) to 10 (really high)?”. Ratings for perceived risk were low <4, intermediate 5–7, and high 8–10.

Questionnaire completion took at least 20 min. All participants received a short description of the objectives of the study and how to use the questionnaire.

If a participant refused to answer or did not know how the answer to a particular question, responses were coded as “don’t know” or “no answer”. These participants were not excluded from the analysis.

### 2.3. Diagnostic Criteria

MRI exams were performed within one week before regularly scheduled blood transfusions using a 1.5T scanner (Signa Artist; GE Healthcare, Milwaukee, WI, USA). For MIO assessment, three parallel short-axis views (basal, medium, and apical) of the left ventricle (LV) were acquired with a gradient-echo multiecho T2* sequence [6,28,31]. Images were analyzed using a previously validated, custom-written software (Hippo-MIOT^®^, Version 2.0, Consiglio Nazionale delle Ricerche and Fondazione Toscana Gabriele Monasterio, Pisa, Italy, Year 2015) [32]. The software provided the T2* value on each of 16 segments of the LV, according to the standard AHA/ACC model [33]. The global heart T2* was obtained by averaging all segmental T2* values, and a value <20 ms was considered indicative of significant MIO [7].

The presence of traditional CVRFs (family history, smoking, alcohol abuse, hypertension, diabetes mellitus, dyslipidemia, obesity) was determined based on patients’ report, physical examination, and laboratory tests. A family history of CVD was defined as a self-reported parental history of heart attack, angina pectoris, or stroke. Smoking habits were self-reported. Alcoholism was defined as the consumption of >60 g of alcohol per day in men and >40 g per day in women. Hypertension was defined as either a former medical history of hypertension, elevated blood pressure measurement (systolic blood pressure ≥ 140 mm Hg or diastolic blood pressure ≥ 90 mm Hg), or the use of blood pressure-lowering drugs. Diabetes mellitus was defined by fasting plasma glucose ≥ 126 mg/dL or 2-h plasma glucose ≥ 200 mg/dL during an oral glucose tolerance test or a random plasma glucose ≥ 200 mg/dL with classic symptoms of hyperglycemia or hyperglycemic crisis [34]. Total cholesterol levels ≥ 200 mg/dL were considered hypercholesterolemia, and triglycerides levels ≥ 150 mg/dL were considered hypertriglyceridemia. Patients taking lipid-lowering drugs were considered patients with dyslipidemia, regardless of the lipid profile. Patient height and weight were used to calculate body mass index (BMI) and obesity was defined as BMI of 30.0 kg/m^2^ or greater.

The following cardiovascular complications were considered: heart failure (HF), arrhythmias, pulmonary hypertension (PH), myo/pericarditis, and myocardial infarction. HF was diagnosed by clinicians based on symptoms, signs, and instrumental findings according to the AHA/ACC guidelines [35]. Arrhythmias were diagnosed if documented by ECG or 24 h Holter ECG and if requiring specific medications. Arrhythmias were classified according to the AHA/ACC guidelines [36]. PH was diagnosed if the trans-tricuspidal velocity jet by echocardiography was greater than 3.2 m/s [37]. In presence of clinical manifestations, the diagnosis of myo/pericarditis required confirmation by cardiac biomarkers (troponin), non-invasive imaging modalities, and biopsy where indicated [38]. Non-fatal myocardial infarction was defined as typical chest pain with elevated cardiac enzyme levels and with or without ST elevation [39].

### 2.4. Statistical Analysis

Data analyses were conducted with the IBM SPSS Statistics 20 statistical package.

Continuous variables were described as mean ± standard deviation (SD). Categorical variables were expressed as frequencies and percentages.

The normality of distribution of the parameters was assessed by using the Kolmogorov–Smirnov test. For continuous values with normal distribution, comparisons between groups were made by independent-samples *t*-test (for 2 groups) or one-way ANOVA (for more than 2 groups). Wilcoxon’s signed rank test and Kruskal–Wallis test were applied for continuous values with non-normal distribution. The Bonferroni adjustment was used in all pairwise comparisons. χ^2^ testing was performed for non-continuous variables.

A 2-tailed probability <0.05 was considered statistically significant.

## 3. Results

### 3.1. Characteristics of Respondents

Male and female patients with TM were comparable for age, age at the start of regular transfusions and chelation, frequency of splenectomy, hemoglobin and serum ferritin levels, and global heart T2* values (Table 1). Moreover, no gender difference was detected in the marital status or level of education.

Eighty-three (59.7%) patients had at least one CVRF; of them, 73.5% had a single CVRF, 21.7% had two CVRFs, and the remaining 4.8% had three or more CVRFs. Among the risk factors, the following frequencies were detected: family history 11.5%, smoking 38.8%, alcohol abuse 2.2%, hypertension 5.0%, diabetes mellitus 16.5%, dyslipidemia 1.4%, obesity 6.5%. No significant association was detected between gender and the presence of CVRFs.

Twenty-five (18.0%) patients had active or prior and resolved cardiovascular diseases: 11 arrhythmias, 11 HF, 1 HF+ arrhythmias, 1 PH, and 1 myocarditis. The prevalence of CVDs was significantly higher in males than in females (26.9% vs. 12.6%; *p* = 0.034). Patients with CVDs were significantly older than patients without CVDs (43.97 ± 7.29 years vs. 40.23 ± 8.06 years; *p* = 0.013) but showed comparable global heart T2* values (36.69 ± 14.28 ms vs. 39.05 ± 8.38 ms; *p* = 0.600).

### 3.2. Knowledge and Awareness of Cardiovascular Disease Risk

The knowledge and the perception of cardiovascular disease risk among TM patients are summarized in Table 2.

The first two questions were multiple-choice questions. A total of 59.7% of patients named tumors in general as the biggest health problem for people of their age and gender while a smaller group of patients (37.4%) named tumors as the greatest danger to their health in the future. In the two aforementioned questions, a significantly higher percentage of females than males selected tumors in general. Cardiovascular diseases were designated as the greatest danger to their health in the future by a significantly higher percentage of males than females. There was no association between the selection of CVDs as the biggest future danger and age, presence of CVRFs, and a positive CVD history.

The other questions were simple yes/no questions. Only the 44.4% of patients were aware that CVDs are not primarily a male illness. There were no sex differences in relation to perceptions of cardiovascular disease as a leading cause of death.

No significant differences emerged according to marital status or educational level. However, although the statistical significance was not reached, patients with the lowest educational level tended to indicate cardiovascular diseases as the greatest danger to their future health less frequently than patients with high school or university-level education (29.7% vs. 44.05 and 59.3%; *p* = 0.061).

### 3.3. Knowledge of CVRFs and Preventive Actions

Table 3 presents patient knowledge of the main risk factors and preventive actions. Family history was the less-frequently recognized CVRF (77.1%) while overweight/obesity was recognized by all patients. Of the preventive actions, the most commonly cited was engaging in physical activity (95.6%).

No significant gender differences were found in the knowledge of CVRFs and preventive measures for CVDs. Having a CVRF was not associated with better perceptions of the main risk factors and preventive actions. With the exclusion of the family history, the level of education did not influence the knowledge of CVRFs.

### 3.4. Self-Rated Health

The average of SRH scores of the TM patients was 6.47 ± 1.57 (on a 1–10 scale). SRH scores were not significantly associated with gender, age, marital status, education level, presence of CVRFs or CVD, and significant MIO (Table 4).

According to our three-category scale for scoring SRH, the 8.0% of TM patients rated their health as poor, 64.2% as acceptable, and 27.8% as good.

### 3.5. Perception of CVD Risk

Among the 130 patients who answered the question about the personal CVD risk, 36 (27.7%) considered themselves to be at low risk, 65 (50.0%) at intermediate risk, and 29 (22.3%) at high risk.

The perceived risk was not influenced by gender, age, marital status, education level, presence of at least one CVRF, and a positive history of CVDs (Table 5). Global heart T2* values were comparable among the groups identified based on the perceived risk, but no patient with a low perceived CVD risk had a pathological global heart T2* value.

Patients with a low perceived CVD risk reported a significantly higher SRH score than patients with moderate and high perceived CVD risk (*p* = 0.006 and *p* = 0.003, respectively).

## 4. Discussion

Adequate awareness of cardiovascular disease is a prerequisite for the adoption of preventive actions and healthy behaviors, and it is particularly important for TM patients due to the heavy burden of CVD, which is associated with multiple causes (myocardial iron overload, traditional cardiovascular risk factors, cardiovascular inflammation, and fibrosis) [10,11,12,13,14]. This is the first study evaluating gender differences in knowledge and awareness of CVDs, knowledge of CVRFs and preventive measures, and perception of health status among Italian patients with TM.

Although almost the 60% of patients named tumors in general as the biggest health problem for people of their age and gender, a considerably lower percentage of patients (37.4%) indicated tumors as the greatest danger to their health in the future. TM patients seemed to perceive themselves as being at higher risk for cardiovascular complications in comparison to the general population. Awareness of CVDs as the major future health issue was significantly lower amongst female patients. Our finding is in line with that of a recent cross-sectional telephone survey involving 2609 individuals without chronic diseases from six European countries [40]. In general, women attribute much less importance to heart disease than it warrants. Indeed, more than half of our patients still belief that CVDs are primarily a male illness. Moreover, in Italy TM patients are well informed about their prognoses and it is probable that they know that in TM the male gender is associated with a significantly higher prevalence of cardiovascular diseases [3,24,25,27].

Our TM patients showed a good knowledge of the CVDs’ main risk factors and of measures for reducing the risk of getting CVDs. Overweight/obesity was recognized by all patients, and it can be postulated that the widespread attention to the aesthetic aspect typical of our era contributes to this awareness. Surprisingly, despite the high prevalence, the clear association with tissue iron overload, and the recognized value as a CVD equivalent, diabetes was one of the most underestimated CVRFs. These data are in accordance with previous surveys on the general population [21,41]. The knowledge of CVRFs was not influenced by gender and, unexpectedly, by the presence of CVRFs, not supporting the general belief that subjects with medical conditions have a greater understanding of the risk associated with these conditions. The educational level did not emerge as a major determinant of knowledge of cardiovascular disease risk. Conversely, different studies conducted on the general population demonstrated a strong association between higher education and better health literacy [21,40]. The most likely explanation of our finding is that among the TM population, the frequent doctor–patient encounters represent the most important source of medical information, compensating for education-induced biases.

Self-rated health, called also “perceived” or “subjective” health, is a subjective indicator of health status, measuring a person’s perception of his/her overall health [42]. Only 8.0% of our TM patients gave a score lower than 4 (on a scale from 0 to 10) to their current health status. This finding confirms the good health status of TM patients in Italy. Thanks to the adequate transfusion regimen, the high majority of our patients had a hemoglobin level within the target range and, of consequence, a strong improvement of the anemia-related symptoms such as fatigue, general weakness, and decreased mental alertness. Moreover, in Italy, as in most high-income countries, safe blood, oral iron chelators, noninvasive techniques for iron overload assessment, and new therapies for the eradication of chronic hepatitis C have generated improved patient outcomes and well-being.

In our study population, SRH was not influenced by gender, age, marital or educational status, iron levels, and history of heart disease, in contrast with other studies [43]. Floris et al., in their cohort of 190 TM patients found a significantly higher health perception among females and patients without cardiac diseases. Important differences between the two studies may account for the discrepancy in the results. All patients in the abovementioned study were followed by a single thalassemia center, located in the Sardinia Island, while we recruited patients from different regions of Italy, obtaining a more representative cohort of the thalassemic Italian population. It is not clear how Floris et al., defined the cardiac disease and there is no information about its prevalence in the whole study population and in both males and females. Finally, the scales used to rate the health status were different (5 vs. 10 points).

It should be pointed out that, although frequently done in epidemiological research, the crude comparison of SRH between males and females may be biased by the different attitudes toward evaluating SRH by gender. While males tend to reflect mainly serious and life-threatening diseases and think of good health more in terms of physical health (bodily activity or function), women tend to be more inclusive, taking into account not only serious and life-threatening diseases, but also mild symptoms, chronic conditions, feelings, and health activities [44,45].

The assessment of the perceived CVD risk can provide an insight into how TM patients see CVD, thereby facilitating the development of effective interventions for CVD prevention. Our data showed that less than one-quarter of TM patients perceived themselves as being at high risk. Although a large percentage of participants had CVD risk factors, the perception of their own CVD risk was virtually the same as the perception of those who had no such CVD risk factors. This datum seems to suggest that, although TM patients had satisfactory general knowledge about cardiovascular risk factors and prevention, they consider CVRFs to play a significantly smaller role in their own lives. Conversely, MIO emerged as an important CV risk determinant for the patients, since no patient with a low perceived CVD risk had a pathological global heart T2* value. This finding could be explained by the fact that for a long time MIO was perceived as the prevalent CVD risk factor in the TM population. So, the fact that the frequency of MIO was not significantly different among males and females is the most plausible explanation for the absence of an association between gender and perceived CV risk. Moreover, it may be speculated that a role may also have been played by the optimism bias, a cognitive bias leading people to perceive their risk to be lower than their peers. Optimism bias can arise from either underestimating one’s own risk or overestimating the risk of the average person; people believe that good things are more likely to happen to themselves than to average others, and bad things are more likely to happen to others [46]. Previous research has shown that men are more optimistic than women regarding different issues, including the risks to health [47,48,49].

## 5. Conclusions

Our study highlighted the need to increase the awareness of CVD as the greatest health issue, especially among the female TM population. TM patients showed good levels of knowledge of the CVDs’ main risk factors, without a gender difference, but they seemed to underestimate the impact of CVRFs on the occurrence of cardiovascular events in their own life. Future educational programs are necessary to inform TM patients about the different actors, besides MIO, that play a role in the development of cardiovascular complications, thus fostering a more accurate risk perception.

## Figures and Tables

**Table 1 jcm-11-03736-t001:** Comparison of demographics and health characteristics between male and female respondents.

	Males (N = 52)	Females (N = 87)	*p*-Value
**Age (years)**	40.88 ± 7.87	40.91 ± 8.17	0.791
**Marital status, N (%)**			0.972
** single**	23 (44.2)	40 (46.0)	
** married/living togheter**	25 (48.1)	41 (47.1)	
** divorced/separated**	4 (7.7)	6 (6.9)	
** widowed**	0 (0.0)	0 (0.0)	
**Education, N (%)**			0.490
** lower secondary school or less**	11 (21.2)	26 (29.9)	
** high school**	31 (59.6)	44 (50.6)	
** university**	10 (19.2)	17 (19.5)	
**Regular transfusion starting age (months)**	15.20 ± 7.92	14.41 ± 13.82	0.123
**Chelation starting age (years)**	4.33 ± 3.32	3.53 ± 3.06	0.201
**Splenectomy, N (%)**	27 (51.9)	43 (49.4)	0.776
**Pre-transfusion hemoglobin (g/dL)**	9.73 ± 0.42	9.79 ± 0.49	0.997
**Mean serum ferritin (ng/mL)**	817.29 ± 585.01	1049.79 ± 1084.58	0.717
**Global heart T2* values (ms)**	39.46 ± 9.04	38.12 ± 10.07	0.331
**Global heart T2* < 20 ms, N (%)**	3 (5.8)	7 (8.0)	0.615
**Cardiovascular risk factors N (%)**	32 (61.5)	51 (58.6)	0.734
**History of cardiovascular diseases, N (%)**	14 (26.9)	11 (12.6)	0.034

N = number.

**Table 2 jcm-11-03736-t002:** Knowledge and awareness of CVD among Italian TM patients according to gender.

	Males (N = 52)	Females (N = 87)	*p*-Value
**Biggest health problem for people of your age and gender, N (%)**			
Tumors (in general)	25 (48.1)	58 (66.7)	0.031
Cardiovascular diseases	17 (32.7)	19 (21.8)	0.158
Diabetes	10 (19.2)	13 (14.9)	0.510
Hepatic tumors	9 (17.3)	10 (11.5)	0.334
**Greatest danger to your health in the future, N (%)**			
Tumors (in general)	14 (26.9)	38 (43.7)	0.048
Cardiovascular diseases	28 (53.8)	32 (36.8)	0.049
Diabetes	15 (28.8)	22 (25.3)	0.646
Hepatic tumors	9 (17.3)	13 (14.9)	0.712
**Cardiac disorders develop gradually over the years and are often not recognized**			
Yes	49/50 (78.0)	73/84 (86.9)	0.178
**Cardiovascular diseases are an almost exclusively male condition**			
No	20/47 (42.6)	36/79 (45.6)	0.742
**Cardiovascular diseases are the main cause of death in Italy**			
Yes	36/50 (72.0)	48/80 (60.0)	0.164
**Cardiovascular diseases cause more victims than breast/prostate cancer in Italy**			
Yes	35/48 (72.9)	47/77 (61.0)	0.174

N = number.

**Table 3 jcm-11-03736-t003:** Patient’s knowledge about main cardiovascular risk factors and preventive measures for cardiovascular diseases, according to gender, presence of CVRFs, and education level.

	Males (N = 52)	Females (N = 87)	*p*-Value	No CVRF (N = 56)	CVRF (N = 83)	*p*-Value	Lower Secondary School or Less (N = 37)	High School (N = 75)	University (N = 27)	*p*-Value
**Cardiovascular risk factors**										
**Family history**	40/50 (80.0)	61/81 (75.3)	0.535	42/52 (80.8)	59/79 (74.7)	0.417	15/32 (53.1)	62/73 (84.9)	22/26 (84.6)	0.001
**Smoking**	49/51 (96.1)	84/85 (98.8)	0.556	53/55 (96.4)	80/81 (98.8)	0.565	34/34 (100)	73/75 (97.3)	26/27 (96.3)	0.571
**Hypertension**	48/51 (94.1)	81/83 (97.6)	0.368	53/54 (98.1)	76/80 (95.0)	0.648	32/35 (91.4)	71/72 (98.6)	26/27 (96.3)	0.184
**Diabetes mellitus**	39/48 (81.3)	67/79 (84.8)	0.601	44/53 (83.0)	62/74 (83.8)	0.909	25/33 (75.8)	58/69 (84.1)	23/25 (92.0)	0.252
**Dyslipidaemia**	50/51 (98.0)	82/85 (96.5)	1.000	53/55 (96.4)	79/81 (97.5)	1.000	32/35 (91.4)	73/74 (98.6)	27/27 (100.0)	0.069
**Overweight/Obesity**	51/51 (100.0)	86/86 (100.0)	1.000	55/55 (100.0)	82/82 (100.0)	1.000	35/35 (100.0)	75/75 (100.0)	27/27 (100.0)	1.000
**Preventive actions for cardiovascular diseases**										
**Quit smoking**	44/51 (86.3)	68/84 (81.0)	0.425	44/54 (81.5)	68/81 (84.0)	0.709	25/35 (77.1)	62/74 (83.8)	23/26 (88.5)	0.489
**Have blood pressure checked**	38/48 (79.2)	75/83 (90.4)	0.073	45/54 (83.3)	68/77 (88.3)	0.415	29/33 (87.9)	60/72 (83.3)	24/26 (92.3)	0.498
**Keep glycaemia under control**	46/51 (90.2)	67/75 (89.3)	0.876	44/50 (88.0)	69/76 (90.8)	0.615	27/30 (90.0)	64/72 (88.9)	22/24 (91.7)	0.926
**Physical activity**	49/52 (94.2)	81/84 (96.4)	0.544	53/55 (96.4)	77/81 (95.1)	1.000	34/37 (91.9)	69/72 (95.8)	27/27 (100.0)	0.293
**Lose weight**	48/50 (96.0)	72/81 (88.9)	0.204	47/53 (88.7)	73/78 (93.6)	0.321	29/32 (90.6)	67/73 (91.8)	24/26 (92.3)	0.971

N = number, CVRF = cardiovascular risk factor.

**Table 4 jcm-11-03736-t004:** SRH status of the subgroups with different demographic and health status.

	Average of SRH Scores	*p*-Value
**Gender**		0.124
** male**	6.75 ± 1.28	
** female**	6.29 ± 1.71	
**Age**		0.415
** <30 years**	7.00 ± 1.52	
** 30–40 years**	6.62 ± 1.65	
** 40–50 years**	6.30 ± 1.42	
** ≥50 years**	6.33 ± 2.06	
**Marital status**		0.956
** single**	6.43 ± 1.77	
** married/living togheter**	6.48 ± 1.41	
** divorced/separated**	6.60 ± 1.43	
**Education**		0.448
** lower secondary school or less**	6.25 ± 1.78	
** high school**	6.59 ± 1.57	
** university**	6.41 ± 1.28	
**Serum ferritin**		0.581
** <1000 ng/mL**	6.58 ± 1.49	
** ≥** **1000 ng/mL**	6.37 ± 1.88	
**CVRFs**		0.113
** no**	6.73 ± 1.54	
** yes**	6.28 ± 1.58	
**History of CVDs**		0.733
** no**	6.46 ± 1.63	
** yes**	6.48 ± 1.33	
**Global heart T2***		0.893
** ≥20 ms**	6.46 ± 1.55	
** <20 ms**	6.60 ± 1.89	

SRH = self-rated health, CVRF = cardiovascular risk factor, CVD = cardiovascular disease.

**Table 5 jcm-11-03736-t005:** Perceived heart disease risk versus demographics and clinical characteristics.

	Perceived CVD Risk			*p*-Value
	Low (N = 36)	Intermediate (N = 65)	High (N = 29)	
**Females, N** **(%)**	20 (55.6)	43 (66.2)	17 (58.6)	0.540
**Age** **(years)**	40.73 ± 9.35	41.35 ± 6.33	41.02 ± 9.02	0.948
**Marital status, N (%)**				0.697
** single**	17 (47.2)	26 (40.0)	14 (48.3)	
** married/living togheter**	15 (41.7)	34 (52.3)	14 (48.3)	
** divorced/separated**	4 (11.1)	5 (7.7)	1 (3.4)	
**Education, N (%)**				0.517
** lower secondary school or less**	12 (33.3)	13 (20.0)	7 (24.1)	
** high school**	19 (52.8)	36 (55.4)	17 (58.6)	
** university**	5 (13.9)	16 (24.6)	5 (17.2)	
**Mean serum ferritin** **(ng/mL)**	819.97 ± 671.11	935.62 ± 840.39	1317.81 ± 1582.44	0.478
**CVRFs, N** **(%)**	24 (66.7)	35 (53.8)	18 (62.1)	0.427
**History of CVDs, N** **(%)**	3 (8.3)	14 (21.5)	7 (24.1)	0.175
**Global heart T2* values** **(ms)**	40.97 ± 4.03	39.16 ± 10.14	36.74 ± 11.69	0.586
**Global heart T2* <** **20 ms, N** **(%)**	0 (0.0)	7 (10.8)	3 (10.3)	0.125
**SRH score**	7.19 ± 1.31	6.29 ± 1.40	5.90 ± 1.84	0.001

CVD = cardiovascular disease, N = number, CVRF = cardiovascular risk factor, SRH = self-rated health.

## Data Availability

The data presented in this study are available on request from the corresponding author. The data are not publicly available due to privacy.

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
