# Peer review of "Gender Differences in Knowledge and Perception of Cardiovascular Disease among Italian Thalassemia Major Patients"

_jcm, 2022, doi:10.3390/jcm11133736_

Round 1
Reviewer 1 Report
This study aimed to evaluate the gender differences in knowledge and perception of cardiovascular disease among Italian thalassemia major patients.
There are minor concerns related to this manuscript to be published in Journal of Clinical Medicine.
#1: Item 2.1 entitled Study design, there is no description of the type of study, please include it.
#2: Was any sample calculation performed for this study?
#3: Absence of the description of the captions and analyzes in table 1. Acctually, “Table 1” looks like a Chart. Please, correct it. (Same for “table” 2, 3, 4 and 5).
Author Response
This study aimed to evaluate the gender differences in knowledge and perception of cardiovascular disease among Italian thalassemia major patients.
There are minor concerns related to this manuscript to be published in Journal of Clinical Medicine.
A: We would like to thank the Reviewer for the encouraging feedback and constructive critique and for the effort regarding this manuscript. We have addressed each of the raised concerns, which have substantially improved the manuscript.
#1: Item 2.1 entitled Study design, there is no description of the type of study, please include it.
A: We have now clarified in the text that this is a cross-sectional survey.
#2: Was any sample calculation performed for this study?
A: We did not perform a-priori power analysis.
#3: Absence of the description of the captions and analyzes in table 1. Acctually, “Table 1” looks like a Chart. Please, correct it. (Same for “table” 2, 3, 4 and 5).
A: Captions have now been added in all Tables.
Title of Table 1 has been changed as follows: “Comparison of demographics and health characteristics between male and female respondents.”.
Title of Table 2 has been changed as follows: “Knowledge and awareness of CVD among Italian TM patients according to gender.”.
Reviewer 2 Report
The concept of the study is interesting; however, the paper's presentation quality can be improved. There are some typos, which should be corrected. Moreover, the study needs some revision of critical points concerning the methodology, data presentation, and interpretation.
Abstract:
- Line 28: Add the percentage of female participants.
- To make the abstract more scientific, it is recommended to bring some crucial statistics (results) and their significance level.
Introduction:
- General comment: well written.
- Only write the study hypothesis after the aim of the study. What was your research question?
Materials and Methods:
- Line 92: Add the percentage of female participants.
- Lines 96-97: The number of people who were asked to participate in the study and did not agree is necessary to estimate the selection bias.
- I recommend drawing participants' flowcharts.
- Does the study have an ethics code or a registry?
- Line 102: The questionnaire used should be further explained. Have they been validated? Are they reliable?
Results:
The analyses are not comprehensive. Confounding factors have not been well identified and have not been controlled in analyses.
Very basic analyses were done. Knowledge is a complex and multifactorial concept requiring rigorous/sophisticated analysis.
For example, what was the level of education of the study participants? Marital status? Income? Social status?
At least the educational status of the participants influences the results.
Discussion
I was not convinced by the discussion of the article.
Author Response
The concept of the study is interesting; however, the paper's presentation quality can be improved. There are some typos, which should be corrected. Moreover, the study needs some revision of critical points concerning the methodology, data presentation, and interpretation.
A: We would like to thank the Reviewer for the encouraging feedback and constructive critique and for the effort regarding this manuscript. We have addressed each of the raised concerns, which have substantially improved the manuscript.
Abstract:
- Line 28: Add the percentage of female participants.
A: The percentage of female participants (62.7%) has now been added.
- To make the abstract more scientific, it is recommended to bring some crucial statistics (results) and their significance level.
A: We have now specified the significant differences between males and females.
Introduction:
- General comment: well written.
A: We thank the Reviewer for this positive feedback.
- Only write the study hypothesis after the aim of the study. What was your research question?
A: Our aim was to evaluate the gender differences in knowledge and perception of cardiovascular disease among Italian thalassemia major patients, without any a-priori hypothesis.
Materials and Methods:
- Line 92: Add the percentage of female participants.
A: The percentage of female participants (62.7%) has now been added.
- Lines 96-97: The number of people who were asked to participate in the study and did not agree is necessary to estimate the selection bias.
A: As now specified in the text, all patients agreed to participate.
- I recommend drawing participants' flowcharts.
A: As now specified in the text, all patients agreed to participate.
- Does the study have an ethics code or a registry?
A: All patients were enrolled in the E-MIOT project, approved by the Institutional Ethics Committee of Area Vasta Nord Ovest (protocol code 56664, date of approval October 8, 2015). Whitin this project, a clinical data registry, easily accessible via a web interface, has been created.
- Line 102: The questionnaire used should be further explained. Have they been validated? Are they reliable?
A: As now clearly stated, the questionnaire was previously validated in the general population [ref]. Moreover, we have now better described some parts of the survey.
Results:
The analyses are not comprehensive. Confounding factors have not been well identified and have not been controlled in analyses.
Very basic analyses were done. Knowledge is a complex and multifactorial concept requiring rigorous/sophisticated analysis.
For example, what was the level of education of the study participants? Marital status? Income? Social status?
At least the educational status of the participants influences the results.
A: We strongly agree with the Reviewer that many factors may influence the knowledge of CVD risk. So, following the Reviewer’s suggestion, we have now taken into account in all our analyses also marital status and education levels (see Tables and Results).
In our TM population marital status did not influence knowledge of CVDs and CVRFs or preventive measures and self-assessed health status. Surprisingly, the educational level did not emerge as a major determinant of knowledge of cardiovascular disease risk. Conversely, different studies conducted on the general population demonstrated an strong association between higher education and better health literacy (ref). The most likely explanation of our finding is that among the TM population the frequent doctor–patient encounters represent the most important source of medical information, compensating for education-induced biases
Discussion
I was not convinced by the discussion of the article.
A: The Discussion has been modified.
Round 2
Reviewer 2 Report
Thanks to the authors.
Almost all of my comments/concerns have been addressed, and I have no other comments/concerns.